# Prechtl's method to assess general movements: Inter-rater reliability during the preterm period

Angélica Valencia[1,2]*, Carlos Viñals[3]°, Elsa Alvarado[3]°, Marcela Balderas[3]°, Joëlle Provasi[2]

1 Faculty of Psychology, Universidad Cooperativa, Cali, Colombia, 2 Cognitions Humaine et Artificielle -EPHE-PSL, CHArt Laboratory, Aubervilliers, France, 3 Cerebral Palsy Department, Instituto Nacional de Rehabilitación: Luis Guillermo Ibarra Ibarra, México City, México

° These authors contributed equally to this work.
* anmvdiaz@gmail.com

**Data Availability Statement:** All relevant data are within the manuscript and its Supporting Information files.

**Funding:** The author received no specific funding for this work.

## Abstract

### Introduction

Prechtl's method (GMA) is a test for the functional assessment of the young nervous system. It involves a global and a detailed assessment of the general movements (GMs) and has demonstrated validity. Data on the reliability of both assessments in the preterm period are scarce. This study aimed to evaluate the inter-rater reliability for the global and detailed assessments of the preterm writhing GMA.

### Materials and methods

The study participants were 69 infants born at <37 gestational weeks and admitted to the neonatal intensive care unit. They were randomly assigned to five pairs of raters. Raters assessed infants' GMs using preterm videos. Outcome variables were (a) the GMs classification (normal versus abnormal; normal versus abnormal subcategories) and (b) the general movements optimality score (GMOS), obtained through the global and detailed assessments. The Gwet's AC1 and the intraclass correlation coefficient (ICC) were calculated for the GMs classification and the GMOS, respectively.

### Results

The global assessment presented an AC1 = 0.84 [95% CI = 0.54,1] for the GMs binary classification and an AC1 = 0.67 [95% CI = 0.38,0.89] for the GMs classification with abnormal subcategories. The detailed assessment presented an ICC = 0.72 [95% CI = 0.39,0.90] for the GMOS.

### Conclusions

Inter-rater reliability was high and substantial for the global assessment and good for the detailed assessment. However, the small sample size limited the precision of these estimates. Future research should involve larger samples of preterm infants to improve

**Competing interests:** The authors have declared that no competing interests exist.

estimate precision. Challenging items such as assessing the neck and trunk, poor repertoire GMs, and tremulous movements may impact the preterm writhing GMA's inter-rater reliability. Therefore, ongoing training and calibration among raters is necessary. Further investigation in clinical settings can enhance our understanding of the preterm writhing GMA's reliability.

## Introduction

Preterm infants are at risk of neurodevelopmental disorders [1]. Thus, international guidelines recommend performing appropriate neurodevelopmental assessments to identify infants who could benefit from early intervention [2]. Prechtl's method (GMA) is a video-based test for the functional assessment of the young nervous system. The GMA has high clinical utility for premature infants because it consists of an observational assessment of the quality of general movements (GMs). GMs originate endogenously in the entire body in a sequence of spontaneous movements [3]. Variability, complexity, and fluidity are the characteristics that determine the quality of GMs and reflect the integrity of the young nervous system. These characteristics of GMs can be negatively affected by brain abnormalities [4, 5] thus becoming a neurodevelopmental marker [6]. The GMA facilitates the evaluation of GMs in three different developmental periods. The preterm writhing GMA (before term age) and writhing GMA (from term age) identify various neurodevelopmental disorders, including functional, motor, and cognitive issues [7, 8]. The fidgety GMA (from 9 weeks post-term) mainly detects cerebral palsy and cognitive disorders [9, 10].

The three GMA periods include both (a) a global and (b) a detailed assessment of the GMs [11]. During the preterm writhing GMA and writhing GMA, the global assessment involves analyzing the quality of GMs and classifying them as either normal or abnormal. The detailed assessment consists of scoring the GMs' characteristics in the limbs to obtain the GMs optimality score (GMOS). During the fidgety GMA, the global assessment classifies GMs as present, abnormal, or absent, while the detailed assessment scores the motricity and posture to obtain the motor optimality score (MOS).

Clinicians, therapists, and neuroscientists worldwide have received training from the GMs Trust group in using both the global and detailed assessments of the GMA. As a result, studies in clinical and research contexts have evaluated the GMA's validity and reliability in preterm infants. Validity and reliability are psychometric properties that express the appropriateness of a test. Validity indicates the scoring accuracy, and reliability shows its consistency. Inter-rater reliability evaluates scoring consistency when different raters assess the same patients with the same test, and agreement is the degree to which scoring is identical [12]. Studies found that the preterm writhing GMA presents high validity (94%) [13] predicting neurodevelopment in preterm infants and high inter-rater agreement (90%) [14]. Specifically, the inter-rater reliability of the global assessment was high (k > 0.80) for preterm writhing GMA and writhing GMA [15], from moderate (k = 0.50) to substantial (k = 0.80) for preterm writhing GMA and writhing GMA in combined analyzes [16, 17], and from substantial (k = 0.64) to high (k = 0.92) for fidgety GMA [15, 18, 19]. A recent study evaluated the reliability of a checklist for guiding raters during the global assessment. That study found that the inter-rater reliability values (k = 0.68–0.80) for the preterm writhing GMA and writhing GMA were similar to those in previous studies [16]. While limited published data exist on the reliability of detailed assessment, studies have report excellent inter-rater reliability (ICC > 0.87) for both the fidgety

MOS and the fidgety MOS-revised [19, 20]. However, to our knowledge, data on inter-rater reliability for detailed assessment in the preterm writhing period are scarce.

Preterm writhing GMA inter-rater reliability needs to be identified because most studies either focus on fidgety GMA, combine preterm writhing GMA and writhing GMA analyses, or do not evaluate the detailed assessment. Furthermore, there is a scarcity of data regarding the inter-rater reliability for the global and detailed assessments of the preterm writhing GMA within the same sample of preterm infants. This gap highlights the need for studies to complement existing evidence on the inter-rater reliability of the preterm writhing GMA.

Consequently, this study aimed to evaluate the preterm writhing GMA inter-rater reliability for (a) the global and (b) the detailed assessment. Specifically, this study addressed the global assessment because of its utility on preterm infants. It also considered the detailed assessment because studies suggest using it to complement the global assessment [21]. Given its well-documented high validity in predicting later development in high-risk preterm infants [13], understanding the reliability of the preterm writhing GMA is crucial for clinical practice. Establishing this reliability is essential to inform clinicians about the preterm writhing GMA's ability to consistently identify at-risk babies, thus facilitating timely interventions during this critical period of neuroplasticity [22]. According to the hypotheses, inter-rater reliability of the preterm writhing GMA would be high (AC1 > 0.80) for the global assessment and excellent (ICC > 0.75) for the detailed assessment.

## Materials and methods

### Design and setting

This psychometric study on the inter-rater reliability of the preterm writhing GMA is part of a broader prospective longitudinal study on the neurodevelopmental assessment of preterm infants. The infants were recruited between January and November 2017 from the Necker Enfants-Malades and Armand Trousseau University hospitals in France. This research followed the Guidelines for Reporting Reliability and Agreement Studies (GRRAS) [12].

### Ethical approval

This study obtained the approval of the Ethical Committee for the Protection of Persons of Île-de-France V (CPP Ref: d-10-16) and followed the principles set out in the Declaration of Helsinki. The infants' parents were sent an informative letter outlining the study's objectives and methods. In reply, they gave written informed consent to their children's inclusion in the study and the video evaluation of the infant's GMs.

### Participants and raters

The sample size selection complied with Bonett's parameters that recommend 28 participants for reliability studies with five raters and an intraclass correlation coefficient value of 0.80 [23]. The sample was recruited conveniently based on the following inclusion criteria: infants born at <37 gestational weeks and admitted to the neonatal intensive care unit (NICU). Infants with congenital anomalies and a severe illness at the time of GMs assessment were excluded. The study used a computerized randomization method to select pairs of raters from a list of GMA-certified raters (n = 5). According to Table 1, each participant (n = 69) was randomly assigned to be evaluated by a different pair (n = 5) of raters.

**Table 1. Assignment of subjects to be evaluated by each pair of raters.**

| Number of pairs of raters | 1 | 2 | 3 | 4 | 5 | Total |
|---|---|---|---|---|---|---|
| Rater's name | B | D | E | C | B | |
| | C | A | D | E | D | |
| Number of subjects | 13 | 22 | 10 | 4 | 20 | 69 |

## Data collection

The data consisted of the participants' GMs video filmed at <37 corrected weeks before their discharge from hospital. Following the GMA parameters, a remotely controlled video camera was positioned above the infants to capture the entire body. The infants were filmed in the supine position wearing little clothing. The filming began when the infants exhibited spontaneous movements following routine nursing care. Infants of <36 corrected weeks were filmed awake or asleep. Infants of ≥36 corrected weeks were filmed in stage 4 (eyes open, no crying, movement present) [24]. During the filming, interaction with caregivers and movement-limiting objects was avoided. Rater A recorded and edited the videos. She also collected the participants' clinical information from their electronic medical records after completing the GMs assessment.

## General movements assessment

The raters received the videos via an online secure platform with a fifteen-day time frame to complete the GMs assessment. Based on these videos, the raters conducted a global and a detailed GMs assessment according to the preterm writhing GMA criteria [11]. In the global assessment, the raters classified GMs as normal if they exhibited varying speed and spatial occupation of moving extremities, along with complex and fluid articulatory rotations. The raters classified the GMs as abnormal if they showed decreased variability, complexity, and fluency. Abnormal GMs included these four subcategories: (a) poor repertoire if the limbs presented monotonous speed and spatial occupation, (b) cramped synchronized if the limbs presented high rigidity, (c) chaotic if the movements were disorganized and (d) hypokinetic if no GMs were observed. Next, the raters performed the detailed assessment using the scoring sheet to evaluate (from 0 to 2) the following items separately: sequence of the GMs, neck and trunk involvement, superior extremities, and lower extremities. The total of the scores for the items gave the GMOS (from 5 to 42). GMOS was not calculated when GMs were hypokinetic.

The raters were in different countries and did not communicate with each other. The raters were blinded to the participants' assignment, clinical data, and individual identity information (except filming age in corrected weeks) and conducted the evaluations independently. After 45 minutes of viewing the videos, the raters took a 5-minute break to calibrate their perception. They participated in two online previous training sessions, which included studying a preterm writhing GMA pedagogical video and reaching an agreement on two cases.

## Outcome variables

The outcome variables were (a) the GMs classification (normal versus abnormal; normal versus abnormal subcategories) obtained through the global assessment and (b) the GMOS (from 5 to 42) obtained through the detailed assessment.

## Statistical analysis

The data reporting used statistical descriptors according to the level of measurement of the variables, with mean and standard deviation (±SD) for continuous variables, median and interquartile range (IQR) for ordinal variables, and frequency and percentage for nominal variables. The analysis considered two types of GMs classification. Firstly, the binary classification of GMs in normal versus abnormal. Next, the classification of the GMs in normal versus abnormal GMs subcategories (poor repertoire, cramped synchronized, chaotic, or hypokinetic). The Kolmogorov-Smirnov test confirmed the normal distribution of the GMOS.

The study calculated the percentage of agreement and Gwet's AC1 coefficient for the GMs classification to evaluate the inter-rater reliability of the global assessment. The AC1 was used because it corresponds better with the percentage of agreement than the k coefficient and controls the problems associated with the prevalence [25]. Additionally, the AC1 has already been used to assess inter-rater reliability of the writhing GMA in postoperative infants [26]. The interpretation of the AC1 considered fair ($\leq 0.40$), moderate (from 0.41 to 0.60), substantial (from 0.61 to 0.80), and high (from 0.81 to 1.00) inter-rater reliability [27].

The evaluation of the inter-rater reliability of the detailed assessment required the calculation of the intraclass correlation coefficient (ICC) for the GMOS. The model of One-way Random Effect, absolute agreement, and single assessment was used. That model is suitable for designs in which subjects are evaluated by different pairs of randomly selected raters [28]. The interpretation of the ICC considered poor ($\leq 0.40$), fair (from 0.41 to 0.59), good (from 0.60 to 0.74), and excellent (from 0.75 to 1) inter-rater reliability [29]. The standard error of measurement (SEM) was calculated with the formula $SEM = SD/\sqrt{2}$ using the SD of the differences between each pair of raters [30].

Reliability coefficients were calculated for each pair of raters and then averaged to obtain a single inter-rater reliability index. This study considered a two-tailed p-value of $< 0.05$ as significant and calculated 95% confidence intervals (CI). The data analysis was performed using SPSS Statistics version 25.0 and AgreeStat 360 [31].

## Results

### Participants and raters

Eighty-two infants were recruited to ensure sufficient data in the contingency table. Thirteen infants were excluded due to withdrawal (n = 1), congenital disease (n = 1), full-term birth (n = 3), and unavailability for filming before reaching 37 corrected weeks (n = 8). In total, 69 infants were included with 35 (51%) males. Table 2 presents the participants' clinical characteristics. The participants were filmed at 35±1 corrected weeks for an average of 3±1 minutes to obtain 5±1 sequences of GMs.

The raters (n = 5) were three physicians and two psychologists with an average of 18±2 years of experience in child neurodevelopment. Three raters are clinical rehabilitation specialists and the other two work in research. The raters are GMs Trust group certified with 6±5 years of experience in working on the global and detailed assessments of the preterm writhing GMA. The results considered the pairs of raters numbered 1,2,3, and 5. The number 4 pair of raters was excluded due to insufficient participants (n = 4) assignment (see Table 1). Thus, four pairs of raters completed the global and detailed assessments of the GMs in 65 preterm infants during the preterm writhing period.

**Table 2. Clinical characteristics of participants (n = 69).**

| | |
|---|---|
| Gestational age in weeks[a] | 31.6±2 |
| Birth weight in grams[a] | 1543.7±429.8 |
| Head circumference at birth in centimeters[a] | 28.6±2.3 |
| APGAR score at 5 minutes[b] | 10 (4,10) |
| Extremely preterm[c] | 3 (4.3) |
| Extremely low birth weight[c] | 6 (8.7) |
| Small for gestational age[c] | 10 (15.6) |
| Brain ultrasound abnormalities[c] | 29 (46) |
| Intraventricular hemorrhage (IVH)[c] | 12 (19) |
| IVH I[c] | 3 (4.8) |
| IVH II[c] | 6 (9.5) |
| IVH III-IV[c] | 3 (4.8) |
| Periventricular leukomalacia (PVL)[c] | 24 (38.1) |
| PVL I[c] | 18 (28.6) |
| PVL II[c] | 4 (6.3) |
| PVL III-IV[c] | 2 (3.2) |
| Bronchopulmonary dysplasia[c] | 11 (15.9) |

Note: Pediatric specialists, who were blinded to the study, identified morbidities during routine NICU assessments. Small for gestational age was defined as birth weight below the 10th percentile.

Brain ultrasound abnormalities refers to IVH (brain bleeding) and PVL (white matter lesion), graded according to international guidelines [32, 33].

Bronchopulmonary dysplasia was defined as the requirement for oxygen supplementation for 28 or 56 days in very preterm or late preterm infants [34].

[a]Data reported as mean±SD

[b]Data reported as median (min–max, IQR)

[c]Data reported as frequency (percentage)

## Inter-rater reliability for the global assessment

Table 3 shows the distribution of the subjects across GMs categories. The GMs categories presented an agreement percentage of 84% (from 69% to 100%). The category with the highest disagreement among the raters was poor repertoire GMs (from 69% to 75%). None of the GMs categories obtained 100% of agreement among all pairs of raters.

Table 4 presents the inter-rater reliability for GMs classification. The GMs binary classification (normal versus abnormal) obtained an inter-rater agreement percentage of 88% (from 80% to 100%) with a coefficient AC1 = 0.84±0.5 (from 0.68 to 1). The GMs classification with abnormal subcategories (poor repertoire, cramped synchronized, chaotic, or hypokinetic) demonstrated an inter-rater agreement of 72% (from 69% to 77%) and an AC1 = 0.67±0 (from 0.63 to 0.73).

## Inter-rater reliability for the detailed assessment

Table 5 presents the inter-rater reliability for the GMOS. The GMOS obtained an ICC = 0.72 ±8 (from 0.66 to 0.79). The lowest inter-rater reliability was related to the item assessing the neck and trunk involvement with an ICC = 0.44±1 (from 0.19 to 0.66). No GMOS item obtained a perfect inter-rater agreement among all pairs of raters.

**Table 3. Distribution of subjects per pair of raters and category of the GMs evaluated during the preterm period (n = 65).**

| | | Pair of raters 1 (B-C)[a] | | % Agreement | Pair of raters 2 (D-A)[a] | | | % Agreement |
|---|---|---|---|---|---|---|---|---|
| | | Rater C | | | | Rater A | | |
| | Rater B | Yes (Y) | No (N) | | Rater D | Yes (Y) | No (N) | |
| Normal | Yes (Y) | 4 (30) | 0 (0) | 84 | Yes (Y) | 8 (36) | 0 (0) | 90 |
| | No (N) | 2 (15) | 7 (53) | | No (N) | 2 (9) | 12 (54) | |
| Abnormal | Y | 7 (53) | 2 (15) | 84 | Y | 12 (54) | 2 (9) | 90 |
| | N | 0 (0) | 4 (30) | | N | 0 (0) | 8 (36) | |
| Poor repertoire | Y | 4 (30) | 4 (30) | 69 | Y | 8 (36) | 6 (27) | 72 |
| | N | 0 (0) | 5 (38) | | N | 0 (0) | 8 (36) | |
| Cramped synchronized | Y | 1 (7) | 0 (0) | 84 | Y | 0 (0) | 4 (18) | 81 |
| | N | 2 (15) | 10 (76) | | N | 0 (0) | 18 (81) | |
| Chaotic | Y | 0 (0) | 0 (0) | - | Y | 0 (0) | 0 (0) | - |
| | N | 0 (0) | 0 (0) | | N | 0 (0) | 0 (0) | |
| Hypokinetic | Y | 0 (0) | 0 (0) | - | Y | 0 (0) | 0 (0) | - |
| | N | 0 (0) | 0 (0) | | N | 0 (0) | 0 (0) | |
| | | Pair of raters 3 (E-D)[a] | | % Agreement | Pair of raters 5 (D-B)[a] | | | % Agreement |
| | | Rater D | | | | Rater B | | |
| | Rater E | Yes (Y) | No (N) | | Rater D | Yes (Y) | No (N) | |
| Normal | Yes (Y) | 2 (20) | 0 (0) | 100 | Yes (Y) | 3 (15) | 4 (20) | 80 |
| | No (N) | 0 (0) | 8 (80) | | No (N) | 0 (0) | 13 (65) | |
| Abnormal | Y | 8 (80) | 0 (0) | 100 | Y | 13 (65) | 0 (0) | 80 |
| | N | 0 (0) | 2 (20) | | N | 4 (20) | 3 (15) | |
| Poor repertoire | Y | 4 (40) | 3 (30) | 70 | Y | 12 (60) | 1 (5) | 75 |
| | N | 0 (0) | 3 (30) | | N | 4 (20) | 3 (15) | |
| Cramped synchronized | Y | 0 (0) | 3 (30) | 70 | Y | 2 (10) | 0 (0) | 95 |
| | N | 0 (0) | 7 (70) | | N | 1 (5) | 17 (85) | |
| Chaotic | Y | 0 (0) | 0 (0) | - | Y | 0 (0) | 0 (0) | - |
| | N | 0 (0) | 0 (0) | | N | 0 (0) | 0 (0) | |
| Hypokinetic | Y | 1 (10) | 0 (0) | 100 | Y | 0 (0) | 0 (0) | - |
| | N | 0 (0) | 9 (90) | | N | 0 (0) | 0 (0) | |

[a]Data reported as frequency (percentage)

**Table 4. Inter-rater reliability per pair of raters for the GMs classification during the preterm period (n = 65).**

| | Pair of raters 1 (B-C) | | | | Pair of raters 2 (D-A) | | | |
|---|---|---|---|---|---|---|---|---|
| | % Agreement | AC1 | p | CI | % Agreement | AC1 | p | CI |
| Normal versus Abnormal | 84 | 0.70 | 0.001 | [0.28,1] | 90 | 0.82 | 0.000 | [0.57,1] |
| Normal versus Abnormal subcategories | 69 | 0.63 | 0.001 | [0.30,0.69] | 77 | 0.73 | 0.000 | [0.51,0.95] |
| | Pair of raters 3 (E-D) | | | | Pair of raters 5 (D-B) | | | |
| | % Agreement | AC1 | p | CI | % Agreement | AC1 | p | CI |
| Normal versus Abnormal | 100 | 1 | 0.000 | [1,1] | 80 | 0.68 | 0.000 | [0.34,1] |
| Normal versus Abnormal subcategories | 70 | 0.64 | 0.002 | [0.25,1] | 75 | 0.71 | 0.000 | [0.47,0.94] |

AC1, Gwet's reliability coefficient; p, significance level of < 0.05; CI, lower limit, upper limit 95% confidence interval.

**Table 5. Inter-rater reliability for the GMOS by pair of raters during the preterm period (n = 64).**

| | Pair of raters 1 (B-C) | | | | | | Pair of raters 2 (D-A) | | | | | |
|---|---|---|---|---|---|---|---|---|---|---|---|---|
| | Scoring[a] | | ICC | p | CI | SEM | Scoring[a] | | ICC | p | CI | SEM |
| | Rater B | Rater C | | | | | Rater D | Rater A | | | | |
| GMOS | 23.4±7 | 25.8±7 | 0.72 | 0.001 | [0.32,0.90] | 3 | 27±10 | 26.9±9 | 0.79 | 0.000 | [0.56,0.9] | 4 |
| Sequence | 1.2±0 | 1.2±0 | 0.70 | 0.006 | [0.29,0.8] | | 1.2±0 | 1.4±0 | 0.68 | 0.000 | [0.33,0.87] | |
| Neck and trunk | 1.9±0 | 2.6±1 | 0.47 | 0.037 | [0.05,0.86] | | 2.5±0 | 1.9±1 | 0.19 | 0.219 | [0.29,0.60] | |
| Upper extremities | 10.31±3 | 11.92±2 | 0.62 | 0.007 | [0.15,0.86] | | 12.5±3 | 12.5±3 | 0.84 | 0.000 | [0.63,0.94] | |
| Lower extremities | 10±3 | 10±3 | 0.62 | 0.007 | [0.16,0.86] | | 10.8±3 | 12.4±4 | 0.75 | 0.000 | [0.44,0.90] | |
| | Pair of raters 3 (E-D) | | | | | | Pair of raters 5 (D-B) | | | | | |
| | Scoring[a] | | ICC | p | CI | SEM | Scoring[a] | | ICC | p | CI | SEM |
| | Rater E | Rater D | | | | | Rater D | Rater B | | | | |
| GMOS | 23.6±10 | 26.4±6 | 0.73 | 0.005 | [0.23,0.93] | 4 | 26.3±9 | 22.6±7 | 0.66 | 0.000 | [0.32,0.84] | 4 |
| Sequence | 1.1±0 | 1.2±0 | 0.80 | 0.190 | [0.27,0.96] | | 1.2±0 | 1.2±0 | 0.66 | 0.043 | [0.29,0.86] | |
| Neck and trunk | 2.7±1 | 2.5±0 | 0.66 | 0.027 | [0.01,0.93] | | 2.5±0 | 1.9±0 | 0.47 | 0.022 | [0.16,0.76] | |
| Upper extremities | 11.7±4 | 12.5±3 | 0.86 | 0.002 | [0.44,0.97] | | 12.5±3 | 10.31±3 | 0.65 | 0.001 | [0.27,0.85] | |
| Lower extremities | 10.1±5 | 10.8±3 | 0.79 | 0.006 | [0.26,0.96] | | 10.8±3 | 10±3 | 0.55 | 0.007 | [0.12,0.89] | |

Note: The GMOS was not calculated for one subject due to hypokinetic GMs.

[a]Data reported as mean±SD.

ICC, intraclass correlation coefficient; p, significance level of < 0.05; CI, lower limit, upper limit 95% confidence interval; SEM, standard error of measurement.

## Discussion

This study aimed to assess the inter-rater reliability of the preterm writhing GMA, providing evidence on both (a) the global and (b) the detailed assessment of the same sample of preterm infants. We will therefore discuss inter-rater reliability estimates for both the GMs classification and the GMOS, then the precision of these estimates will also be addressed.

### Inter-rater reliability for the global assessment

As expected in our hypothesis, the inter-rater reliability of the global assessment was high (AC1 = 0.84 [95% CI = 0.54,1]) for the GMs binary classification (normal versus abnormal). However, it was substantial (AC1 = 0.67 [95% CI = 0.38,0.89]) for the GMs classification with abnormal subcategories (poor repertoire, cramped synchronized, chaotic, or hypokinetic). Our findings align with prior research on preterm writhing, writhing, and fidgety GMA, which found higher inter-rater reliability for the GMs binary (k > 80) classification compared to the GMs classification with abnormal subcategories (k = 50) [15, 17, 19]. Therefore, we will discuss the factors that influence this disparity in the inter-rater reliability for the global assessment.

The decrease in inter-rater reliability for the GMs classification with abnormal subcategories can be attributed to the nature of reliability coefficients, which tend to diminish when there are more than two classification categories [35]. Additionally, items that are difficult to interpret can impact the inter-rater reliability for the GMs classification with abnormal subcategories [30]. The poor repertoire GMs category could be a challenging item due to the highest bias (0.23) and the highest disagreement (76%) among raters. Although disagreement could have been reduced by using the checklist to guide GMs assessment, our inter-rater reliability estimates agree with those obtained for the checklist (k = 0.68–0.80) [16]. Observations in this study align with previous studies that have suggested the lack of precision of poor repertoire GMs in identifying neurodevelopmental disturbances in preterm infants [36]. Studies have

shown that infants with poor repertoire GMs can transition into normal GMs by the time they reach term age [37]. This imprecision of poor repertoire GMs could also impact inter-rater reliability for the GMs classification with abnormal subcategories.

Therefore, clinical studies recommend combining the binary GMs classification with other neurological measures and neuroimaging to enhance the identification of preterm infants at neurodevelopmental risk [38]. Also, researchers have suggested using the global assessment of the preterm writhing GMA in the framework of longitudinal neurodevelopmental follow-up monitoring [21].

## Inter-rater reliability for the detailed assessment

Contrary to our hypothesis, the detailed assessment demonstrated good inter-rater reliability (ICC = 0.72 [95% CI = 0.35,0.89]) for the GMOS. This observation differs from previous studies that reported higher inter-rater reliability for the fidgety MOS and the fidgety MOS-revised [19, 20]. Therefore, we will now consider the factors that may have influenced the inter-rater reliability value for the GMOS in this study.

Earlier studies have shown that the differences among raters' expertise levels can affect inter-rater reliability for the fidgety GMA [18]. Although the raters have comparable expertise levels in preterm writhing GMA, they come from different professional clinical and research backgrounds. Raters (pair 2) with a research background demonstrated higher inter-rater reliability (ICC = 0.79 [95% CI = 0.56,0.90]) for the GMOS compared to raters from clinical fields. This observation aligns with a study that revealed lower inter-rater reliability for the global assessment of the writhing GMA and fidgety GMA in clinical settings than in research settings [17]. Professional background differences among raters in this study may have influenced inter-rater reliability for the GMOS.

Additionally, the item assessing neck and trunk involvement may be challenging because it presented the highest disagreement (ICC = 0.44 [95% CI = 0.12,0.78]) among raters. The response options for this item make it hard to discriminate between little and no involvement. The item evaluating the presence of tremulous movements may also be challenging because it had the second-highest rater disagreement. Two previous studies could support these observations. One study observed tremulous movements in both normal GMs and abnormal GMs [21] during the preterm writhing and writhing period. The other study demonstrated the clinical imprecision of tremulous movements during the writhing period in identifying neurodevelopmental disturbances in preterm infants [39]. The lack of precision of these challenging items may have impacted the inter-rater reliability for the GMOS.

Given the clinical utility of the detailed assessment, previous studies recommend using it in combination with the global assessment to gain a deeper understanding of specific parameters and trajectories of the GMs in preterm infants [21].

## The precisión of estimates

We also observed wide 95% confidence intervals, which suggest potential limitations in the precision of the inter-rater reliability estimates. Confidence intervals for any inter-rater reliability estimate depend on two factors: sample size and sample variability related to the assessed parameter (in this case, the GMs) [40]. Therefore, we will address how these two factors could have influenced the precision of inter-rater reliability estimates for both the global and the detailed assessments.

Firstly, a small sample size can lead to increased error and increased uncertainty in inter-rater reliability estimates [40]. While our sample size is suitable for reliability studies involving categorical and numerical variables [23, 35], the relatively small number of subjects (n = 69)

might affect the precision of inter-rater reliability estimates for both the global and detailed assessments. Two previous studies with smaller sample sizes reported similar findings. One study (n = 39) on the global assessment of the preterm writhing GMA reported high inter-rater reliability (k > 0.80 [95% CI = 0.40,1]) for the GMs classification but noted a wide 95% confidence interval [15]. Another study (n = 24), focusing on the detailed assessment, reported high inter-rater reliability (ICC = 0.87) for the fidgety MOS but noted a slightly higher measurement error than expected.

Secondly, an increased variability within the sample reduces the precision of inter-rater reliability estimates [41]. The oscillation in the proportion of abnormal GMs (ranging from 53% to 80%) and the standard deviation (25.3±8) of the GMOS may suggest variability within our sample. Thus, the relative heterogeneity related to the GMs within the sample could have influenced the precision of inter-rater reliability estimates for both the global and the detailed assessments. These observations contrast with the findings of a recent study involving a more heterogeneous sample of preterm and term infants with diverse clinical characteristics [20]. In that study, the fidgety MOS-revised exhibited higher inter-rater reliability values (ICC = 0.98 [95% CI = 0.97,0.99]). It is important to note that the study included a significantly larger sample (n = 252).

## Limitations and future research

Our sample size met Bonett's parameters for inter-rater reliability studies [23] but the relatively low number of subjects was a limiting factor for this study. Therefore, future research should consider larger samples of preterm infants to increase the precision of inter-rater reliability estimates for the preterm writhing GMA. While the randomized assignment of participants to randomly formed pairs of raters might minimize bias [28], the convenience recruiting participants from the NICU was another limitation for this study. Participants recruiting in NICU might to explain the high rate of PVL in our sample. Therefore, future studies on high-risk preterm infants could consider recruiting participants upon admission to the NICU to improve the sample's representativeness and the generalizability of inter-rater reliability estimations for the preterm writhing GMA.

## Conclusions

Reliability in identifying preterm infants at neurodevelopmental risk is a critical concern in assessments. This study provides insights into the inter-rater reliability of the preterm writhing GMA for evaluating the functionality of a young nervous system. We observed high and substantial inter-rater reliability for the global assessment, with the binary GMs classification being the most reliable. The detailed assessment showed good inter-rater reliability for the GMOS. However, our small sample size limited the precision of these estimates. Several challenging items, such as assessing neck and trunk involvement, poor repertoire GMs, and tremulous movements contributed to substantial inconsistency among raters. Therefore, ongoing training and rater calibration is necessary to enhance inter-rater reliability for the preterm writhing GMA. The preterm writhing GMA seems to have better inter-rater reliability in research settings than in a clinical environment. Given the utility of the preterm writhing GMA, further investigation in clinical settings is necessary to better understand its inter-rater reliability in identifying preterm infants at a high risk of neurodevelopmental issues.

## Supporting information

**S1 File.**
(XLSX)

## Author Contributions

**Conceptualization:** Angélica Valencia, Joëlle Provasi.

**Data curation:** Angélica Valencia, Carlos Viñals, Elsa Alvarado, Marcela Balderas, Joëlle Provasi.

**Formal analysis:** Angélica Valencia, Joëlle Provasi.

**Investigation:** Angélica Valencia, Carlos Viñals, Elsa Alvarado, Marcela Balderas, Joëlle Provasi.

**Methodology:** Angélica Valencia, Carlos Viñals, Elsa Alvarado, Marcela Balderas, Joëlle Provasi.

**Writing – review & editing:** Angélica Valencia, Joëlle Provasi.

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
