## [Decision Letter · Decision Letter 0]

14 Aug 2023

PONE-D-23-13340Prechtl’s method to assess general movements: Inter-rater reliability for the global and detailed assessment during the preterm periodPLOS ONE

Dear Dr. Angelica VALENCIA

Thank you for submitting your manuscript to PLOS ONE. After careful consideration, we feel that it has merit but does not fully meet PLOS ONE’s publication criteria as it currently stands. Therefore, we invite you to submit a revised version of the manuscript that addresses the points raised during the review process.

Please submit your revised manuscript by  Sep 28 2023 11:59PM. If you will need more time than this to complete your revisions, please reply to this message or contact the journal office at plosone@plos.org. Please include the following items when submitting your revised manuscript:A rebuttal letter that responds to each point raised by the academic editor and reviewer(s). You should upload this letter as a separate file labeled 'Response to Reviewers'.A marked-up copy of your manuscript that highlights changes made to the original version. You should upload this as a separate file labeled 'Revised Manuscript with Track Changes'.An unmarked version of your revised paper without tracked changes. You should upload this as a separate file labeled 'Manuscript'.

We look forward to receiving your revised manuscript.

Kind regards,

Claudia Brogna

Academic Editor

PLOS ONE

Journal Requirements:

Reviewers' comments:

Reviewer's Responses to Questions

**Comments to the Author**

1. Is the manuscript technically sound, and do the data support the conclusions?

Reviewer #1: Partly

Reviewer #2: Partly

2. Has the statistical analysis been performed appropriately and rigorously? 

Reviewer #1: Yes

Reviewer #2: I Don't Know

3. Have the authors made all data underlying the findings in their manuscript fully available?

Reviewer #1: Yes

Reviewer #2: Yes

4. Is the manuscript presented in an intelligible fashion and written in standard English?

Reviewer #1: No

Reviewer #2: Yes

5. Review Comments to the Author

Reviewer #1: This study evaluated GMA global results and GMOS scores for 65 infants videotaped at <37 weeks postmenstrual age prior to hospital discharge. Raters were well qualified to rate videos. Five pairs of raters were enlisted to test the reliability and agreement of the ratings, but one pair of raters only evaluated 4 participants and this pairing was excluded from analysis. Findings were in general agreement with other studies , however the authors identified 95% confidence intervals that included much lower values for reliability. The authors indicated this may affect the generalization of the results. However, the confidence interval would be likely to narrow with larger sample sizes and given their small number of participants, it is not clear if they have proven their conclusion.

Apart from this finding, the results are largely confirming other studies. The discussion is not clearly organized and the conclusions are somewhat vague. It is difficult to understand the main points the authors are trying to make.

Reviewer #2: The paper entitled “Prechtl’s method to assess general movements: Inter-rater reliability for the global and detailed assessment during the preterm period” report on an interesting issue on the use of general movements in preterm newborns.

This paper is of interest for physicians involved in the follow-up of this population of infants. However there are few issues that should be commented.

Introduction

Pag 9 line 57: I’m not sure that the GMs should be considered a “neuromotor test”. It measures the quality of movement detecting disorders of movement (functional limitation), as better explained few lines after.

Results

It seems that the % of abnormal GMs pattern was lower than that of normal GMs. It could be of interest to know more information about the clinical characteristics in term of risk of neurological impairment; for example the US scan data or MRI, seizures, incidence of SGA or other potential risk factors

Discussion

The results are well discussed like as the potential limitations.

Please revise all the manuscript for English

6. PLOS authors have the option to publish the peer review history of their article (what does this mean?). If published, this will include your full peer review and any attached files.

Reviewer #1: No

Reviewer #2: No

---

## [Author Response · Author response to Decision Letter 0]

1 Dec 2023

• Academic Editor: "Please ensure that your manuscript meets PLOS ONE's style requirements, including those for file naming." 

We have revised our manuscript to ensure full compliance with PLOS ONE's style for authors' affiliations, headings, tables, and reference citations. Each file is now named according to the requirements, which includes 'Response to Reviewers', 'Revised Manuscript with Track Changes', 'Manuscript', and 'Supporting Information'.

• Academic Editor: "In your Data Availability statement, you have not specified where the minimal data set underlying the results described in your manuscript can be found. PLOS defines a study's minimal data set as the underlying data used to reach the conclusions drawn in the manuscript and any additional data required to replicate the reported study findings in their entirety. All PLOS journals require that the minimal data set be made fully available."

To address this request, we have uploaded our study’s minimal underlying data set as 'Supporting Information' as an Excel file upon submitting our revised manuscript.

• Reviewer #1: " The discussion is not clearly organized. It is difficult to understand the main points the authors are trying to make."

In response, we have restructured the discussion (page 11, line 262). The discussion now addresses three main results: inter-rater reliability for the global and detailed assessments of general movements (GMs) using the Prechtl' method (GMA) during the preterm period, as well as the precision of these estimates.

Firstly, we addressed the differences in inter-rater reliability for the global assessment (page 12, line 274). Since the GMs classification with abnormal subcategories obtained lower inter-rater reliability compared to the binary GMs classification, we discussed the factors that might have influenced this difference.

Secondly, we discussed the factors that might have influenced the inter-rater reliability of the detailed assessment because the observed inter-rater reliability for the GMOS (GMs optimality score) contradicted our hypothesis and previous research (page 13, line 300). 

Thirdly, we discussed the potential impact of small sample size and variability on the precision of our estimates (page 14, line 326). This was essential because we observed wide 95% confidence intervals associated with our inter-rater reliability index for both the global and detailed assessments.

• Reviewer #1: "The authors identified 95% confidence intervals that included much lower values for reliability. The authors indicated this may affect the generalization of the results. However, the confidence interval would be likely to narrow with larger sample sizes and given their small number of participants, it is not clear if they have proven their conclusion". "The conclusions are somewhat vague. It is difficult to understand the main points the authors are trying to make".

We agree with Reviewer #1 and have updated the conclusion (page 15, line 367). The revised conclusion no longer focuses on problems related to the generalization of results. Instead, it highlights concerns about the precision of estimates due to the wide confidence intervals observed (Gardner and Altman, 1986). The conclusion is now more precise, addressing the potential impact of our small sample size on the width of the confidence intervals, which, in turn, can affect the precision of our inter-rater reliability estimates. Additionally, the conclusion discusses how challenging items and differences among raters might influence inter-rater reliability for the preterm GMA.

• Reviewer #2: "Introduction Pag 9 line 57: I’m not sure that the GMs should be considered a “neuromotor test”. It measures the quality of movement detecting disorders of movement (functional limitation), as better explained few lines after."

We agree with Reviewer #2 and have used the accurate terminology. The revised introduction (page 3, line 64) defines the GMA as a test for the functional assessment of the young nervous system (Einspieler and Prechtl, 2005).

• Reviewer #2: " It could be of interest to know more information about the clinical characteristics in term of risk of neurological impairment; for example, the US scan data or MRI, seizures, incidence of SGA or other potential risk factors" 

In response, we have included an additional table (Table 2 in page 9, line 225) in the results section, which provides information on participants' clinical characteristics in term of risk of neurological impairment. It includes the US scan, incidence of SGA, APGAR at 5 min, extremely low birth weight, and neonatal clinical complications. 

• Reviewer #2: "please revise all the manuscript for English"

In response, we have requested professional language editing services to enhance the quality of English in our manuscript. We uploaded the certification of proofreader.

---

## [Decision Letter · Decision Letter 1]

1 Feb 2024

PONE-D-23-13340R1Prechtl’s method to assess general movements: Inter-rater reliability during the preterm periodPLOS ONE

Dear Dr. VALENCIA,

Thank you for submitting your manuscript to PLOS ONE. After careful consideration, we feel that it has merit but does not fully meet PLOS ONE’s publication criteria as it currently stands. Therefore, we invite you to submit a revised version of the manuscript that addresses the points raised during the review process.

We look forward to receiving your revised manuscript.

Kind regards,

Claudia Brogna

Academic Editor

PLOS ONE

Journal Requirements:

Please review your reference list to ensure that it is complete and correct. If you have cited papers that have been retracted, please include the rationale for doing so in the manuscript text, or remove these references and replace them with relevant current references. Any changes to the reference list should be mentioned in the rebuttal letter that accompanies your revised manuscript. If you need to cite a retracted article, indicate the article’s retracted status in the References list and also include a citation and full reference for the retraction 

Reviewers' comments:

Reviewer's Responses to Questions

**Comments to the Author**

1. If the authors have adequately addressed your comments raised in a previous round of review and you feel that this manuscript is now acceptable for publication, you may indicate that here to bypass the “Comments to the Author” section, enter your conflict of interest statement in the “Confidential to Editor” section, and submit your "Accept" recommendation.

Reviewer #1: (No Response)

Reviewer #2: All comments have been addressed

Reviewer #3: All comments have been addressed

2. Is the manuscript technically sound, and do the data support the conclusions?

Reviewer #1: Yes

Reviewer #2: Yes

Reviewer #3: Yes

3. Has the statistical analysis been performed appropriately and rigorously? 

Reviewer #1: Yes

Reviewer #2: Yes

Reviewer #3: Yes

4. Have the authors made all data underlying the findings in their manuscript fully available?

Reviewer #1: Yes

Reviewer #2: Yes

Reviewer #3: Yes

5. Is the manuscript presented in an intelligible fashion and written in standard English?

Reviewer #1: Yes

Reviewer #2: Yes

Reviewer #3: Yes

6. Review Comments to the Author

Reviewer #1: A table of neonatal morbidities has been added but these should be defined in the text. The rate of PVL seems unusually high and this may reflect the definition of PVL or selection of patients for the study.

Reviewer #2: The authors answered to all the queries. No further comments are needed

Reviewer #3: This study details the inter-rater reliability of preterm writhing GMA global scores as well as the General Movements Optimality Score (GMOS). Strengths of this study include the generalisability of results with raters being in different countries. The small sample size is a limitation, although the authors do very well to explore this in great detail without biasing the results in their favour, rather encouraging the reader to exercise caution and notes for future studies.

Abstract:

Would be good to use the term “preterm writhing GMA” to add clarity about which period of GMs the study focuses on. Similarly, the term “preterm writhing GMA” does not appear anywhere in the abstract. This is important to highlight to differentiate from fidgety GMs.

Introduction:

Ideally throughout the manuscript, the distinction between fidgety and writhing should be made to avoid any confusion. E.g. when discussing previous literature about interrater agreement and predictive validity – are the studies referenced in relation to writhing or fidgety movements? Sometimes it is qualified in the manuscript, and sometimes it is not.

Avoid definite statements, e.g. “…has not been reported”; “this is the first study”. There may be conference abstracts or similar that details this. Instead, consider using “little/scarce publish data exists” or “to our knowledge, few studies report…”/”there is a scarcity of data..”

The authors do well do describe the gap in knowledge (although some expressions could be less definitive, see above), but justification for the importance of the knowing the accuracy in the preterm writhing period is lacking. i.e. what is the clinical benefit of having this knowledge during the preterm period? This point may be able to be addressed easily by highlighting research about preterm writhing movements, specifically, and their relationship with later development (line 92 – unclear if this research refers to preterm writhing or fidgety GMA, best to check).

Methods:

“Crying and hiccupping episodes were suppressed with an online editing program” – I’m not sure I understand how this is possible? Which program was used? Perhaps this program needs to be quoted, as you would with statistical analysis software. This sounds like a very complex procedure and could affect how general movements are observed with human Gestalt perception. Do the authors perhaps mean they were simply edited out?

The authors may want to refer to the work by Dr Crowle Early Human Development, 104, 2017, to further justify the use of Gwets AC1.

Results:

Appropriate, tables clear to read.

Discussion:

Appropriate and well discussed limitations.

7. PLOS authors have the option to publish the peer review history of their article (what does this mean?). If published, this will include your full peer review and any attached files.

Reviewer #1: No

Reviewer #2: No

Reviewer #3: No

---

## [Author Response · Author response to Decision Letter 1]

12 Mar 2024

Academic Editor: " Please review your reference list to ensure that it is complete and correct. If you have cited papers that have been retracted, please include the rationale for doing so in the manuscript text, or remove these references and replace them with relevant current references. Any changes to the reference list should be mentioned in the rebuttal letter that accompanies your revised manuscript. If you need to cite a retracted article, indicate the article’s retracted status in the References list and also include a citation and full reference for the retraction."

We have careful reviewed our reference list. It is complete, accurate, and does not include any retracted papers. We have added four references. 

• Reference 26 (Crowle C, et al., 2017) on page 8, line 201 provides further justification for using the AC1 reliability coefficient, as recommended by Reviewer 3.

• References 32 (Ment LR, et al., 2012), 33 (de Vries L, Eken P, and Dubowitz L, 1992), and 34 (Hadchouel A, and Delacourt C, 2013) on page 9, line 228, define the participants morbidities presented in Table 2, as recommended by Reviewer 1.

Reviewer #1: "A table of neonatal morbidities has been added but these should be defined in the text."

In response, we have presented the criteria for defining the following morbidities: small for gestational age, brain ultrasound abnormalities, intraventricular hemorrhage (IVH), periventricular leukomalacia (PVL), and bronchopulmonary dysplasia (Note in Table 2, page 9, line 228). 

We have removed the prevalence of respiratory distress and infection (Table 2, page 9, line 228) due to missing data regarding the definition of these morbidities. 

Reviewer #1: "The rate of PVL seems unusually high and this may reflect the definition of PVL or selection of patients for the study."

We agree with Reviewer #1 and have added (page 15, line 365) that the high rate of PVL in our sample may be explained by the convenience recruitment of high-risk infants admitted to the NICU, as stated in participants section (page 5, line 141).

Reviewer #3: " Would be good to use the term “preterm writhing GMA” to add clarity about which period of GMs the study focuses on. Similarly, the term “preterm writhing GMA” does not appear anywhere in the abstract. This is important to highlight to differentiate from fidgety GMs". 

"Ideally throughout the manuscript, the distinction between fidgety and writhing should be made to avoid any confusion. E.g. when discussing previous literature about interrater agreement and predictive validity – are the studies referenced in relation to writhing or fidgety movements? Sometimes it is qualified in the manuscript, and sometimes it is not".

We have implemented the suggestion from Reviewer #3 and updated the abstract (page 2, line 35) accordingly. The abstract now uses the term "preterm writhing GMA". This term has been used throughout the entire text to enhance clarity and precision in terminology throughout the manuscript.

Additionally, the updated introduction (page 4, lines 90-95) now distinguishes between writhing GMA and fidgety GMA while summarizing previous literature on interrater agreement and predictive validity.

Reviewer #3: Avoid definite statements, e.g. “…has not been reported”; “this is the first study”. There may be conference abstracts or similar that details this. Instead, consider using “little/scarce publish data exists” or “to our knowledge, few studies report…”/”there is a scarcity of data..”

We have replaced definite statements throughout the entire text, specifically in the abstract (page 2, line 33), introduction (page 4, lines 101 and 106), and discussion (page 11, line 263), using the nuanced statements (e.g., there is a scarcity of data) recommended by Reviewer #3.

Reviewer #3: The authors do well to describe the gap in knowledge (although some expressions could be less definitive, see above), but justification for the importance of the knowing the accuracy in the preterm writhing period is lacking. i.e. what is the clinical benefit of having this knowledge during the preterm period? This point may be able to be addressed easily by highlighting research about preterm writhing movements, specifically, and their relationship with later development (line 92 – unclear if this research refers to preterm writhing or fidgety GMA, best to check).

We agree with Reviewer #3 and have updated the justification (page 4, line 113-118) about the clinical importance of understanding the reliability of the preterm writhing GMA. It now includes the following arguments:

• The demonstrated high validity of the preterm writhing GMA in predicting development in preterm infants (Craciunoiu O, and Holsti L. 2016)

• The value of this information to guide clinicians about consistency of the preterm writhing GMA in identifying at-risk babies and opportunities for early interventions.

Additionally, we have clarified that the validation study cited on page 2, line 92 refers to preterm writhing GMA.

Reviewer #3: "Crying and hiccupping episodes were suppressed with an online editing program” – I’m not sure I understand how this is possible? Which program was used? Perhaps this program needs to be quoted, as you would with statistical analysis software. This sounds like a very complex procedure and could affect how general movements are observed with human Gestalt perception. Do the authors perhaps mean they were simply edited out?"

Yes, we mean that the videos from the participants were edited. Therefore, the updated version of the method section (page 6, line 157) simply states this.

Reviewer #3: "The authors may want to refer to the work by Dr Crowle Early Human development, 104, 2017, to further justify the use of Gwets AC1."

In response, we have included the reference of Crowle C, et al. (2017) in the statistical analysis section (page 8, line 201)

---

## [Editor Report · Decision Letter 2]

25 Mar 2024

Prechtl’s method to assess general movements: Inter-rater reliability during the preterm period

PONE-D-23-13340R2

Dear Dr. Angelica VALENCIA,

We’re pleased to inform you that your manuscript has been judged scientifically suitable for publication and will be formally accepted for publication once it meets all outstanding technical requirements.

Kind regards,

Claudia Brogna

Academic Editor

PLOS ONE
---

## [Editor Report · Acceptance letter]

27 Mar 2024

PONE-D-23-13340R2 

PLOS ONE

Dear Dr. VALENCIA, 

I'm pleased to inform you that your manuscript has been deemed suitable for publication in PLOS ONE. Congratulations! Your manuscript is now being handed over to our production team.

Kind regards, 

on behalf of

Dr. Claudia Brogna 

Academic Editor

PLOS ONE